# MANIFOLD-MATCHING AUTOENCODERS

## ABSTRACT

We propose Manifold-Matching Autoencoders (MMAEs), a simple yet effective framework that aligns autoencoder latent spaces with precomputed geometric references. This is accomplished by using distance-based regularization to match latent and reference distance matrices, enabling the same architecture to achieve different data representations by simply changing the reference embedding. We demonstrate that MMAEs achieve scalable topological control in high-dimensional settings where existing methods become computationally intractable. One key finding is that aligning with PCA yields unexpected benefits: MMAEs achieve SOTA preservation of the original data structure, comparable to sophisticated topological autoencoders, while maintaining significantly better reconstruction quality and more efficient computation. When combining with VAEs, the present regularization has the effect of concentrating variance in fewer dimensions. This balance between structure preservation, variance concentration, and reconstruction fidelity enables superior generative capabilities, including clearer interpolations and more effective discovery of semantically meaningful latent directions for attribute manipulation.

## 1 INTRODUCTION

Autoencoders remain highly relevant today, playing a crucial role in various fields of machine learning and data science. Their ability to efficiently learn compressed representations of data makes them invaluable for tasks such as dimensionality reduction, anomaly detection, denoising, and unsupervised feature learning (Chen & Guo, 2023). However, their lack of ability to preserve the topology or the global structure of the input data, coupled with random weights initialization, can lead to discontinuities in the latent representations that weren't originally present in the input data. These discontinuities can negatively affect the decoder's ability to reconstruct the input (Batson et al., 2021). It is a complex problem as data is usually high-dimensional, where the curse of dimensionality causes distance concentration—pairwise distances become increasingly similar, reducing their discriminative power for capturing meaningful relationships (Aggarwal et al., 2001). This makes direct alignment between latent and input spaces ineffective. Recent approaches address this by incorporating topological data analysis (TDA) (Moor et al., 2020b; Trofimov et al., 2023), which preserves essential structural features (connected components, cycles) through persistent homology rather than preserving all distances indiscriminately. Although these approaches have succeeded in cases for visualization of the bottleneck i.e cases where the bottleneck is 2D/3D, nevertheless, the use of geometric/topological regularizations and more generally manifold learning for learning all-purpose high-dimensional representations (e.g., 128D/256D) remains an open problem (Duque et al., 2023).

Variational autoencoders (VAEs) (Kingma & Welling, 2022) typically require larger bottlenecks, allowing more information to flow through for generating synthetic images, for example. Although the latent space of VAEs is usually understood from a distributional perspective, recent studies have shown that geometric properties of the latent space, such as its shape or density, can help generate better images, an interpretation that extends to other generative models such as GANs (Chadebec & Allassonnière, 2022; Xu et al., 2024). Others advocate that isometric embeddings can help uncover directions or regions that represent the presence or addition/removal of semantically meaningful attributes while maintaining other aspects of an image intact (Kato et al., 2020). Thus, flexibly manipulating latent space geometry is desirable either for preserving the original data topology or for extending useful representations to unseen data for visualization, classification, clustering, or generation.

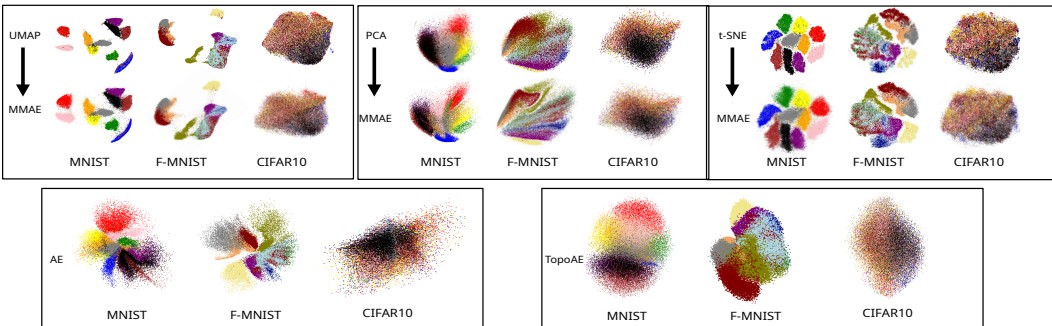

Figure 1: MMAEs copying different 2D representations of the data across three datasets. Standard AE and TopoAE for comparison. Training is unsupervised, classes (colors) are used for visualization purposes only.

We propose a simple distance-based regularization for autoencoders that aligns latent representations with reference embeddings from established dimensionality reduction methods such as PCA (Hotelling, 1933), t-SNE (van der Maaten & Hinton, 2008), and UMAP (McInnes et al., 2018). This approach enables autoencoders to embed new data points within these established geometric structures while maintaining parametric generalization capabilities.

Our method provides distinct advantages across different scenarios. In low-dimensional settings, it transforms autoencoder latent spaces into effective visualization tools that reproduce the structure of classical dimensionality reduction techniques (see Figure 1). In high-dimensional scenarios, it remains competitive with state-of-the-art topological variants (Moor et al., 2020b) while offering reduced computational complexity. Additionally, for generative applications with VAEs, our regularization produces a clear variance hierarchy within the dimensions of the bottleneck, in contrast to the traditional VAE, a useful property for tasks such as semantic interpolation and attribute manipulation (Kato et al., 2020).

We identify two critical factors for optimal generative performance: achieving high absolute variance values in the latent space and concentrating this variance across fewer dimensions. Our experiments reveal that geometric regularization choice significantly impacts both factors, enabling MMVAEs to outperform other approaches through clearer images and better separation of variation factors. For example, in 3DShapes (Burgess & Kim, 2018) we achieve single-attribute manipulation (shape, orientation, size) via linear latent directions. In CelebA (Liu et al., 2015), we similarly add or remove smiles, or other attributes, without altering other facial features. This represents important progress toward controllable image synthesis.

## 2 BACKGROUND: GEOMETRY, TOPOLOGY, & MANIFOLD LEARNING

Geometry in latent spaces concerns quantitative metric properties (distances, angles, coordinates), while topology refers to structural properties (connectivity, clustering, manifold structure) that remain invariant under continuous deformations. Multiple geometries can share identical topology—circles and ellipses have different geometries but the same topological structure. The importance of understanding data topology has been recognized since the 1960s, including by Rosenblatt, the inventor of the perceptron (Rosenblatt, 1962). Topology is fundamentally tied to the manifold hypothesis underlying modern dimensionality reduction: high-dimensional data $\mathbf{X} = \{\mathbf{x}_i\}_{i=1}^{k}$ with $\mathbf{x}_i \in \mathbb{R}^n$ typically lies on or near a lower-dimensional manifold $\mathcal{M} \subset \mathbb{R}^n$. Classical approaches like PCA (Hotelling, 1933; Jolliffe, 2002) find linear subspaces maximizing variance through $\mathbf{X} \approx \sum_{i=1}^{l} a_i \mathbf{v}_i$, providing computationally efficient representations that preserve the global variance structure, which is beneficial for tasks requiring interpretable features and approximate data reconstruction. Modern nonlinear methods address PCA's limitations in capturing curved manifolds. UMAP (McInnes et al., 2018) preserves the structure of the local neighborhood through fuzzy topological representations, producing embeddings that maintain the cluster relationships essential for classification and clustering tasks. t-SNE (van der Maaten & Hinton, 2008) optimizes local pairwise similarities via probability distributions, excelling at revealing local structure for visualization and exploratory

analysis. These dimensionality reduction methods provide intermediate representations that make high-dimensional relationships computationally tractable and geometrically interpretable—reducing the curse of dimensionality while preserving task-relevant structure.

Standard autoencoders (Hinton & Salakhutdinov, 2006) lack explicit topological constraints, potentially mapping similar input points to distant latent regions, creating discontinuities that affect downstream applications (Batson et al., 2021). Recent topological AE variants (Moor et al., 2020b; Chen et al., 2022; Trofimov et al., 2023) incorporate regularization using distance matrices $\boldsymbol{D}^X$ (input space) and $\boldsymbol{D}^Z$ (latent space) through persistence homology. However, computational complexity scales poorly with dimensionality (Moor et al., 2020a). An alternative is to directly manipulate the geometry of the latent space. This idea considers that by defining a specific geometry, the desired topology can be consequently achieved. However, for synthetic data, it is easier to define a useful geometry as the manifold is known, but for the real-world scenarios this is not the case. Additionally, as dimensionality of the bottleneck grows, imposing a geometry through direct alignment of coordinates becomes significantly more intractable (Duque et al., 2023), making these approaches ineffective in high-dimensional scenarios.

Multidimensional scaling (MDS) (Torgerson, 1952) provides a classical approach that reconstructs coordinates directly from pairwise distance matrices. The key insight is that while points $\mathbf{x}_i, \mathbf{x}_j \in \mathbb{R}^n$ may have many coordinates, their Euclidean distance $d_{ij} = \|\mathbf{x}_i - \mathbf{x}_j\|_2$ reduces their relationship to a single scalar value. Remarkably, collecting all such pairwise distances into a matrix $\mathbf{D}$ contains sufficient information to recover the original geometric configuration. Classical MDS formalizes this by converting distance relationships into geometric configurations through eigendecomposition of the associated Gram matrix (Borg & Groenen, 2005; Schoenberg, 1935). We will see that Manifold-Matching Autoencoders, by aligning the latent space to a known geometry through pairwise distances, and minimization of the MMLoss, are able to approximate reference geometric configurations.

## 3 MANIFOLD-MATCHING AUTOENCODERS

### 3.1 FORMULATION

Manifold-Matching Autoencoders (MMAE) extend vanilla autoencoders by adjusting their latent space shape to match that of precomputed embeddings $\boldsymbol{E} = \{\boldsymbol{e}_i\}_{i=1}^k$ with $\boldsymbol{e}_i \in \mathbb{R}^l$ ($l < n$) of the input data $\boldsymbol{X} = \{\boldsymbol{x}_i\}_{i=1}^k$ with $\boldsymbol{x}_i \in \mathbb{R}^n$, where these embeddings are obtained via a mapping $u : \mathcal{X} \to \mathcal{E}$. The key insight is to transfer the topological structure captured by pairwise distances in $\mathcal{E}$ to constrain the latent space $\mathcal{Z}$ through regularization. See Figure 2 for a visual overview of the approach.

MMAEs. like other autoencoders, is the composition of an encoder $g_{\boldsymbol{\theta}} : \mathcal{X} \to \mathcal{Z}$ that maps input data to a latent representation $\boldsymbol{z}_i = g_{\boldsymbol{\theta}}(\boldsymbol{x}_i) \in \mathbb{R}^m$ ($m < n$), and a decoder $h_{\varphi} : \mathcal{Z} \to \mathcal{X}$ that reconstructs the input as $\hat{\boldsymbol{x}}_i = h_{\varphi}(\boldsymbol{z}_i)$. MMAEs, however, use a training objective that combines reconstruction fidelity with topological structure preservation:

$$\mathcal{L}(\boldsymbol{\theta}, \varphi; \boldsymbol{X}, \boldsymbol{E}) = \mathcal{L}_r(\boldsymbol{X}, \hat{\boldsymbol{X}}) + \lambda \mathcal{L}_{mm}(\boldsymbol{Z}, \boldsymbol{E}) \tag{1}$$

where $\mathcal{L}_r$ is a reconstruction loss (e.g., Mean Squared Error), $\mathcal{L}_{mm}$ is our manifold matching loss, and $\lambda \in \mathbb{R}^+$ is a weighting parameter controlling the regularization strength.

### 3.2 MANIFOLD-MATCHING LOSS (MM LOSS)

Given batch $\boldsymbol{X} \in \mathbb{R}^{p \times n}$ with latent representation $\boldsymbol{Z} = g_{\boldsymbol{\theta}}(\boldsymbol{X}) \in \mathbb{R}^{p \times m}$ and embeddings $\boldsymbol{E} \in \mathbb{R}^{p \times l}$, we define pairwise distance matrices $\boldsymbol{D}^E, \boldsymbol{D}^Z \in \mathbb{R}^{p \times p}$ with entries:

$$d_{ij}^E = \|\boldsymbol{e}_i - \boldsymbol{e}_j\|^2, \quad d_{ij}^Z = \|\boldsymbol{z}_i - \boldsymbol{z}_j\|^2, \quad \tilde{d}_{ij}^E = \frac{d_{ij}^E}{\|\boldsymbol{D}^E\|_F}, \quad \tilde{d}_{ij}^Z = \frac{d_{ij}^Z}{\|\boldsymbol{D}^Z\|_F} \tag{2}$$

Where $\tilde{d}_{ij}^E, \tilde{d}_{ij}^Z$ are normalized distances and $\| \cdot \|_F$ denotes the Frobenius norm. The manifold matching loss is:

$$\mathcal{L}_{mm}(\boldsymbol{Z}, \boldsymbol{E}) = \frac{1}{p^2} \|\tilde{\boldsymbol{D}}^E - \tilde{\boldsymbol{D}}^Z\|_F^2 \tag{3}$$

The MMAE optimization problem can be formulated as:

$$(\boldsymbol{\theta}^*, \varphi^*) = \arg\min_{\boldsymbol{\theta}, \varphi} \mathbb{E}_{\boldsymbol{X} \sim p_{\text{data}}} \left[ \mathcal{L}_r(\boldsymbol{X}, h_\varphi(g_{\boldsymbol{\theta}}(\boldsymbol{X}))) + \lambda \mathcal{L}_{mm}(g_{\boldsymbol{\theta}}(\boldsymbol{X}), \boldsymbol{E}) \right] \qquad (4)$$

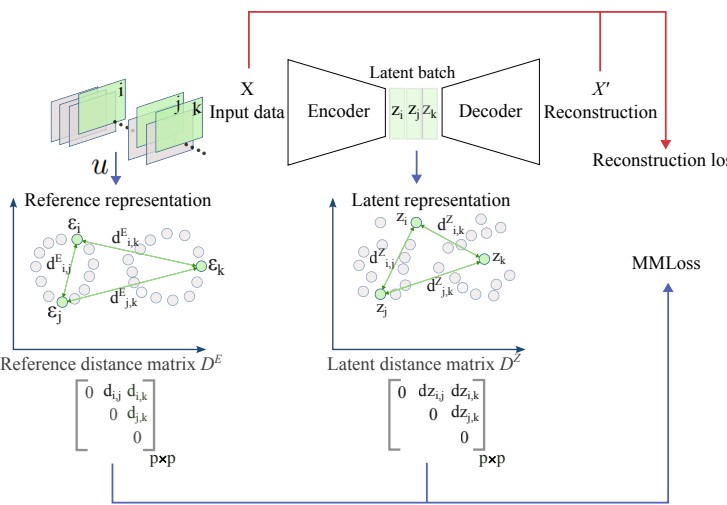

Figure 2: Overview of the current approach. One fundamental insight is that distances in the reference representation take into consideration all of the dataset points, thus incorporating into each training batch knowledge about the global structure of the full embedded dataset.

While computing distance matrices appears computationally intensive, the complexity $\mathcal{O}(p^2)$ is typically manageable for reasonably sized batches. Moreover, since the dimensionality of the spaces in which we compute distances ($m$ and $l$) is substantially lower than the input dimensionality $n$, the computational overhead remains tractable. See Algorithm 1 for the implementation.

---

**Algorithm 1** MMAE Training Procedure

---

**Require:** Dataset $\boldsymbol{X}$, precomputed embeddings $\boldsymbol{E}$, encoder $g_{\boldsymbol{\theta}}$, decoder $h_\varphi$
 1: **for** each mini-batch $\boldsymbol{X}_b, \boldsymbol{E}_b$ in $\boldsymbol{X}, \boldsymbol{E}$ **do**
 2:     Compute latent codes $\boldsymbol{Z}_b = g_{\boldsymbol{\theta}}(\boldsymbol{X}_b)$
 3:     Compute reconstructions $\hat{\boldsymbol{X}}_b = h_\varphi(\boldsymbol{Z}_b)$
 4:     Compute reconstruction loss $\mathcal{L}_r = \|\boldsymbol{X}_b - \hat{\boldsymbol{X}}_b\|^2$
 5:     Compute distance matrices $\boldsymbol{D}^E$ and $\boldsymbol{D}^Z$
 6:     Compute normalized distances $\tilde{\boldsymbol{D}}^E$ and $\tilde{\boldsymbol{D}}^Z$
 7:     Compute manifold matching loss $\mathcal{L}_{mm}$ using Equation (4)
 8:     Compute total loss $\mathcal{L} = \mathcal{L}_r + \lambda \mathcal{L}_{mm}$
 9:     Update parameters $\boldsymbol{\theta}$ and $\varphi$ using gradient descent on $\mathcal{L}$
10: **end for**

---

### 3.3 COMPARISON TO MDS

Distance preservation provides a principled approach to geometric alignment. When our manifold-matching loss minimizes $\mathcal{L}_{mm}(\mathbf{Z}, \mathbf{E}) = \frac{1}{p^2}\|\tilde{\mathbf{D}}^E - \tilde{\mathbf{D}}^Z\|_F^2$, it drives the latent space $\mathbf{Z}$ to preserve the same pairwise distance relationships as the reference embedding $\mathcal{L}_{mm} \to 0 \Rightarrow \tilde{\mathbf{D}}^Z \approx \tilde{\mathbf{D}}^E$. This establishes that our neural encoder $g_\theta$ learns a parametric extension of classical MDS—enabling generalization to new data points while preserving the geometric structure captured by the reference embedding method. The key advantage is dimensionality flexibility: our distance-based approach works when $m \neq l$ (e.g., 256D latent matching 2D or 100D reference), as normalized distances are scale-invariant and independent of ambient dimensionality.

## 4 RELATED WORK

Moor et al. (2020b) propose Topological Autoencoders (TopoAEs) that preserve topological structures via persistent homology using regularization:

$$\mathcal{L} := \mathcal{L}_r + \lambda \mathcal{L}_t \tag{5}$$

where $\mathcal{L}_t$ is the topological loss and $\lambda$ controls regularization strength. Like our method, TopoAEs operate on distance matrices $\mathbf{D^X}$ and $\mathbf{D^Z}$ from input and latent spaces. However, rather than preserving all pairwise distances, TopoAEs use persistent homology to identify and select only topologically significant distances through persistence pairings $\Pi_{\mathbf{X}}$ and $\Pi_{\mathbf{Z}}$. These pairings act as filters that extract subsets of distances $\mathbf{D}_\Pi^{\mathbf{X}}$ and $\mathbf{D}_\Pi^{\mathbf{Z}}$ corresponding to edges that create or destroy topological features (e.g., the edge that closes a loop). The topological loss then matches these filtered distance sets between spaces. While this selective approach targets the most structurally important relationships, it requires complex persistent homology computation. Moor et al. (2020a) investigate alternative distance metrics but find limited benefits, while subsequent topology-aware methods (Chen et al., 2022; Trofimov et al., 2023) inherit similar computational limitations.

Duque et al. (2023) propose Geometry Regularized Autoencoders (GRAEs) with a fundamentally different approach: rather than computing topological structure during training, they rely on pre-computed reference embeddings where the topological knowledge is externalized to the reference algorithm:

$$\mathcal{L} := \mathcal{L}_r + \lambda \mathcal{L}_g, \quad \mathcal{L}_g = \sum_{i=1}^{k} \|\boldsymbol{\epsilon}_i - g_\theta(\mathbf{x}_i)\|^2 \tag{6}$$

where $\mathcal{L}_g$ enforces coordinate-wise alignment between latent representations and reference embeddings $\mathbf{E} = \{\boldsymbol{\epsilon}_i\}_{i=1}^{k}$ computed using UMAP (McInnes et al., 2018) or PHATE (Moon et al., 2019). GRAE aims to exactly reproduce reference coordinates through direct alignment, requiring identical dimensionality between reference and latent spaces. In contrast, our approach preserves relative distances between points rather than absolute coordinates. The normalization in our manifold-matching loss enables the autoencoder to scale its representation freely, so long as normalized relative distances are maintained. While GRAE forces exact coordinate reproduction, our distance-based regularization acts as a geometric "compass" that guides the latent space organization without constraining absolute positioning or scale.

## 5 EXPERIMENTS

### 5.1 SETTINGS:

**Datasets:** We use three simple real-world image datasets **MNIST**, **Fashion-MNIST** ($28 \times 28 \times 1$) (Lecun et al., 1998; Xiao et al., 2017), and **CIFAR10** ($32 \times 32 \times 3$) (Krizhevsky, 2009). In the generative scenarios we explore the **dSprites** ($64 \times 64$)(Matthey et al., 2017), **3DShapes** ($64 \times 64 \times 3$) (Burgess & Kim, 2018) and **CelebA** ($256 \times 256 \times 3$) (Liu et al., 2015).

**Models & Training:** In the case of CIFAR10, MNIST, and F-MNIST we use a simple MLP autoencoder based on the DeepAE architecture proposed by (Moor et al., 2020b). In the generative cases, we use convolutional layers. Details are given in A.3. All models employ batch normalization and ADAM optimizer (Kingma & Ba, 2017). Models are trained at most for 30 epochs. Reference mechanisms used are PCA (Hotelling, 1933), and UMAP (McInnes et al., 2018) (t-SNE (van der Maaten & Hinton, 2008) is limited to at most 3D for visualization purposes only). In the high-dimensional scenarios, the embeddings used have at most 100 components in MMAE (in the CelebA case, for example, this means that we use $100 \div (256 \times 256 \times 3) \approx 0.051\%$ of the original data dimensionality), while GRAE requires the embeddings to have the same dimensionality as the bottlenecks. Latent dimensionality in all cases is 256D.

### 5.2 EXPERIMENT: DIMENSIONALITY REDUCTION QUALITY

For large bottlenecks, the original data topology can be measured by how well relative distances and neighborhoods are preserved when moving from one space to another. In this case, our models are

evaluated comparing the latent spaces $\mathcal{Z}$ to the input space $\mathcal{X}$. **Trustworthiness** and **Continuity** (Venna & Kaski, 2001) quantify local neighborhood preservation by measuring how well $k$-nearest neighbor relationships are maintained across spaces, where trustworthiness penalizes false neighbors appearing in $\mathcal{Z}$ that were not neighbors in $\mathcal{X}$, while continuity penalizes true neighbors from $\mathcal{X}$ that are separated in $\mathcal{Z}$, both yielding values in [0,1] with higher scores indicating better local structure; we average the results for different values $k = [3; 5; 10; 25; 100]$. We call the product of $Trust \times Cont$ "*neighborhood preservation*" for better visualization. **RMSE** captures global geometric consistency by comparing normalized pairwise distance matrices between spaces (not related to reconstruction error); and **MRRE** (Lee et al., 2007) assesses global topological preservation through rank correlation analysis of distance orderings.

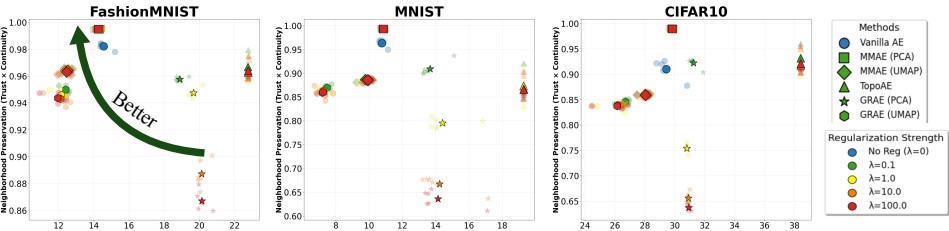

Figure 3: NLDR quality metrics for varying regularization strength $\lambda$. Comparison between MMAE, GRAE, TopoAE, and AE. Five runs per model.

**Results:** Figure 3 shows that MMAE (PCA) consistently outperforms all other approaches in preserving local neighborhoods, as measured by the product (Trust.$\times$ Cont.) on the y-axis, and maintaining comparable RMSE performance to the standard AE. TopoAE exhibits the highest RMSE, which remains unchanged across regularization strengths—a characteristic consistent with the authors' findings (Moor et al., 2020b). The UMAP variants achieved the lowest RMSE across all datasets, which is consistent with UMAP's focus on global structure preservation with better performance in clustering (see A.4.1). TopoAE and MMAE (PCA) also showed better robustness to different types of noise, as seen in A.4.2. In contrast, UMAP variants achieved the best clustering likely due to the more advanced capabilities of this dimensionality reduction tool to separate classes (as can be seen for the 2D case of MNIST and F-MNIST in Figure 1).

### 5.3 EXPERIMENT: SEMANTIC INTERPOLATIONS

In 3DShapes and dSprites, the attribute vectors are computed using a mean latent difference approach. For each factor $f$ and value $v$, we first compute the mean latent representation $\boldsymbol{\mu}_f(v) = \frac{1}{N_v} \sum_{i=1}^{N_v} g_{\boldsymbol{\theta}}(\mathbf{x}_i)$, where $\mathbf{x}_i \in \mathcal{X}$ are samples with factor $f$ equal to value $v$, $g_{\boldsymbol{\theta}} : \mathcal{X} \to \mathcal{Z}$ is the encoder function, and $N_v$ is the number of such samples. The direction vectors $\mathbf{d} \in \mathbb{R}^m$ are then defined as differences between these mean latents: for discrete factors like shape transformations, $\mathbf{d}_{\text{shape}} = \boldsymbol{\mu}_{\text{shape}}(v_{\text{target}}) - \boldsymbol{\mu}_{\text{shape}}(v_{\text{source}})$, while for continuous factors like color or scale, $\mathbf{d}_{\text{factor}} = \boldsymbol{\mu}_f(v_{\text{max}}) - \boldsymbol{\mu}_f(v_{\text{min}})$. Interpolation is performed by modifying the original latent code $\mathbf{z}_i = g_{\boldsymbol{\theta}}(\mathbf{x}_i)$ as $\mathbf{z}_i' = \mathbf{z}_i + \alpha\mathbf{d}$, where $\alpha \in \mathbb{R}$ controls the interpolation strength. For CelebA, the procedure is the same, however, labels have binary values (0 or 1), with no intermediate values for "smiling", "blond hair" or other attributes.

**Results:** MMVAE (PCA) consistently achieves superior semantic interpolation across datasets. In 3DShapes (Figure 5), it successfully manipulates individual attributes (shape, orientation, scale) without affecting others (color, background), while competing approaches exhibit factor entanglement (**a); d); e)**). Approaches using UMAP (**c); f)**) show second best approach but become more unstable for extreme values of $\alpha$. For CelebA, MMVAE (PCA) produces higher-quality interpolations with better-formed details and attribute-specific control compared to standard VAE and GRVAE. The key advantage lies in variance distribution: both topological regularization and our approach concentrate variance in fewer dimensions, but MMVAE (PCA) achieves substantially higher absolute maximum variance values ($\approx 4.0$ for 3DShapes, $\approx 20.0$ for CelebA) compared to TopoVAE's severely limited values ($\approx 0.015$ in both datasets) (Figure 7). GRVAE fails to achieve similar concentration despite coordinate alignment. This variance concentration advantage correlates with

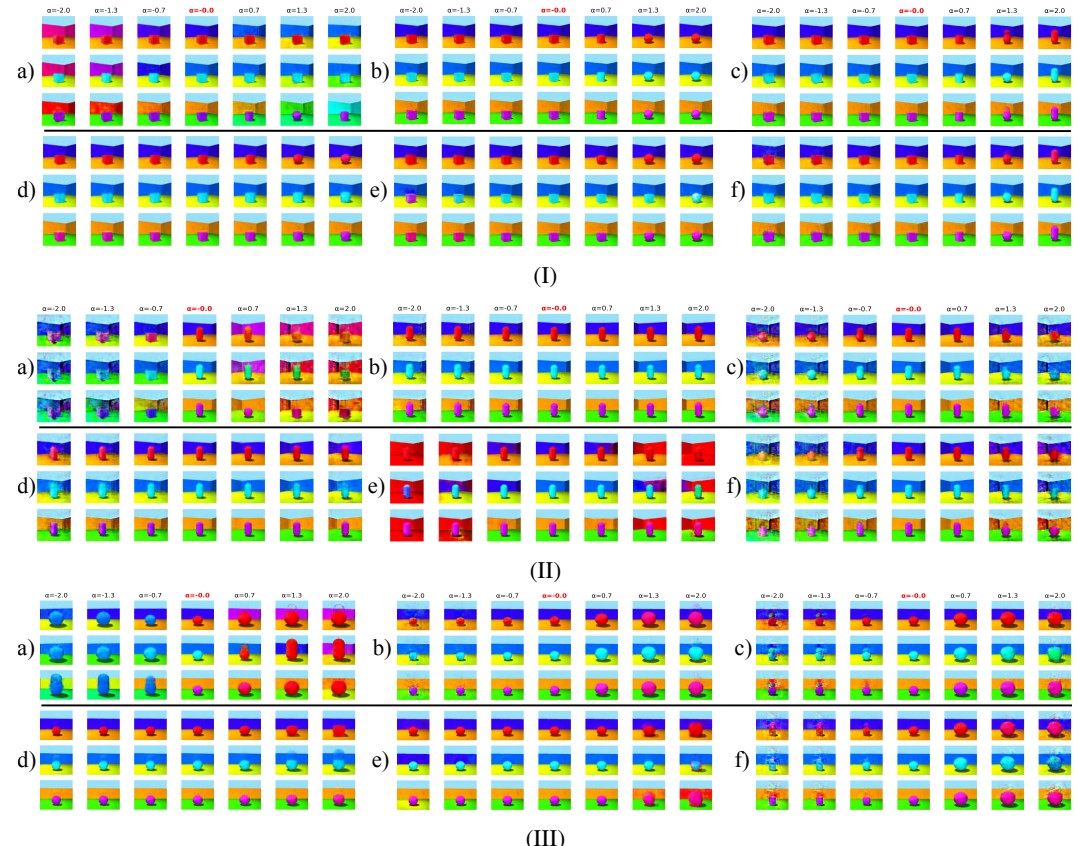

Figure 4: 3DShapes interpolation results. I Cube to sphere transformation, II pill orientation changes, III sphere scaling variations. **a)** Standard VAE, **b)** MMAE (PCA), **c)** MMAE (UMAP), **d)** TopoVAE, **e)** GRVAE (PCA), **f)** GRVAE (UMAP). $\alpha \in [-2; 2]$.

neighborhood preservation results (Figure 6I), where only MMVAE-PCA and TopoVAE maintain strong performance at larger scales k, with MMVAE-PCA providing superior reconstruction quality. GRVAE consistently degrades at higher k values, confirming that variance concentration is essential for preserving local geometric structure.

## 6 DISCUSSION

Manipulating latent space geometry offers significant benefits beyond 2D/3D visualization, but remains challenging for real-world datasets with unknown structure. Our MMAE method offers a flexible way to take advantage of the richness of AEs while guiding the shape of latent space via potentially simple methods. For example MMAE (PCA) produces a latent space with effective structural bias for image datasets, measured through NLDR quality metrics. The generative variant, MMVAE, also preserved this structural bias and produced distinctive variance patterns characterized by concentration in fewer dimensions and higher absolute values. In contrast to the more uniform variance distribution observed in standard VAEs ($\approx 1.0$). This PCA-like hierarchical structure facilitates clearer interpretation (Kato et al., 2020; Casella et al., 2022; Pham et al., 2022) and supports effective linear interpolation. Following Bengio et al. (2013)'s point of view of a flat manifold, where linear latent interpolations yield smooth output transitions, VAE manifolds naturally exhibit minimal curvature (Shao et al., 2018). We hypothesize that our distance-based regularization enhances this property by preserving the geometric relationships from well-structured reference embeddings, resulting in more accurate attribute manipulation. Mathieu et al. (2019) introduce "decomposition", a generalization of disentanglement (Locatello et al., 2019), characterized by two requirements: (1) appropriate latent overlap controlled by encoding stochasticity (the $\beta$ parameter

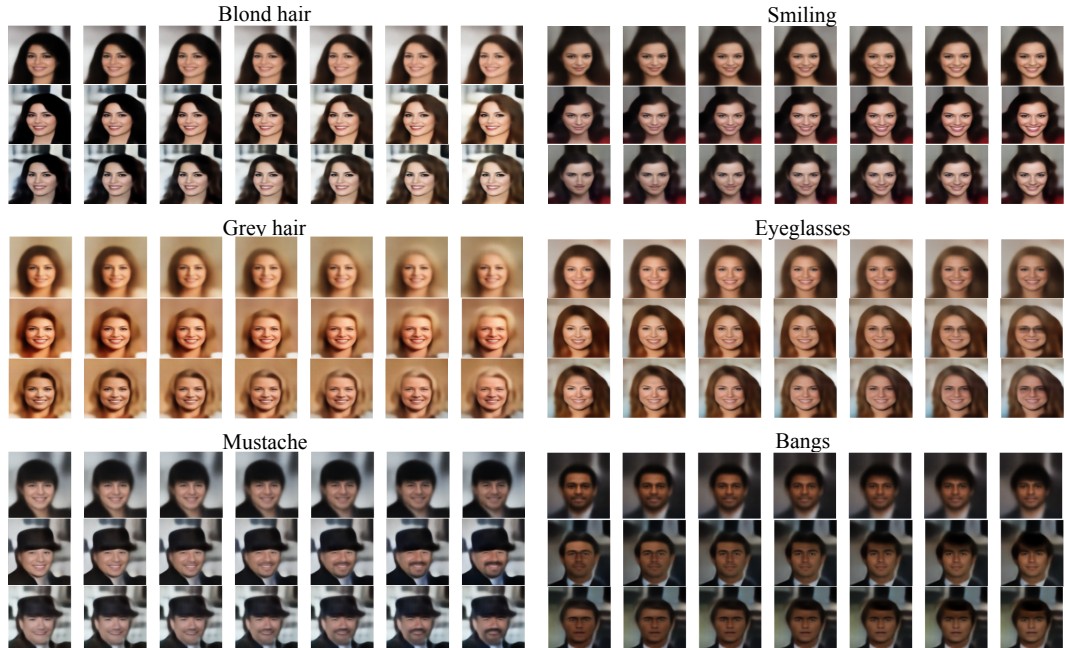

Figure 5: CelebA interpolations for $\alpha \in [-1; 1]$. **Top:** Standard VAE, **Middle:** MMVAE (PCA), **Bottom:** GRVAE (PCA).

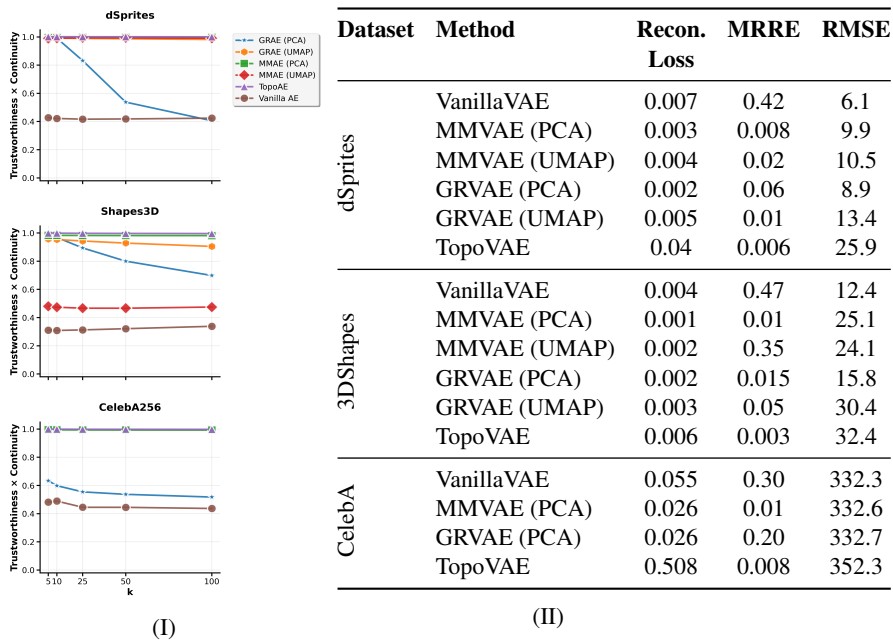

(I)

| Dataset | Method | Recon. Loss | MRRE | RMSE |
|---------|--------|-------------|------|------|
| dSprites | VanillaVAE | 0.007 | 0.42 | 6.1 |
| | MMVAE (PCA) | 0.003 | 0.008 | 9.9 |
| | MMVAE (UMAP) | 0.004 | 0.02 | 10.5 |
| | GRVAE (PCA) | 0.002 | 0.06 | 8.9 |
| | GRVAE (UMAP) | 0.005 | 0.01 | 13.4 |
| | TopoVAE | 0.04 | 0.006 | 25.9 |
| 3DShapes | VanillaVAE | 0.004 | 0.47 | 12.4 |
| | MMVAE (PCA) | 0.001 | 0.01 | 25.1 |
| | MMVAE (UMAP) | 0.002 | 0.35 | 24.1 |
| | GRVAE (PCA) | 0.002 | 0.015 | 15.8 |
| | GRVAE (UMAP) | 0.003 | 0.05 | 30.4 |
| | TopoVAE | 0.006 | 0.003 | 32.4 |
| CelebA | VanillaVAE | 0.055 | 0.30 | 332.3 |
| | MMVAE (PCA) | 0.026 | 0.01 | 332.6 |
| | GRVAE (PCA) | 0.026 | 0.20 | 332.7 |
| | TopoVAE | 0.508 | 0.008 | 352.3 |

(II)

Figure 6: I NLDR quality metrics (Trustworthiness × Continuity) across different values of k for VAE variants on three datasets. II Quantitative performance metrics including reconstruction loss, MRRE, and RMSE for the same models and datasets.

in $\beta$-VAE), and (2) the aggregate posterior $q(\mathbf{z})$ matching a prior that encodes desired dependency structures among latent variables. We interpret the alignment of pairwise distances via $\mathcal{L}_{mm}$ in VAEs as imposing a geometric prior—where the reference embedding's structure (PCA, UMAP, t-SNE) defines the desired geometric organization. Our results support this interpretation, as distance-regularized models consistently outperform standard VAEs in controlled manipulation of individual

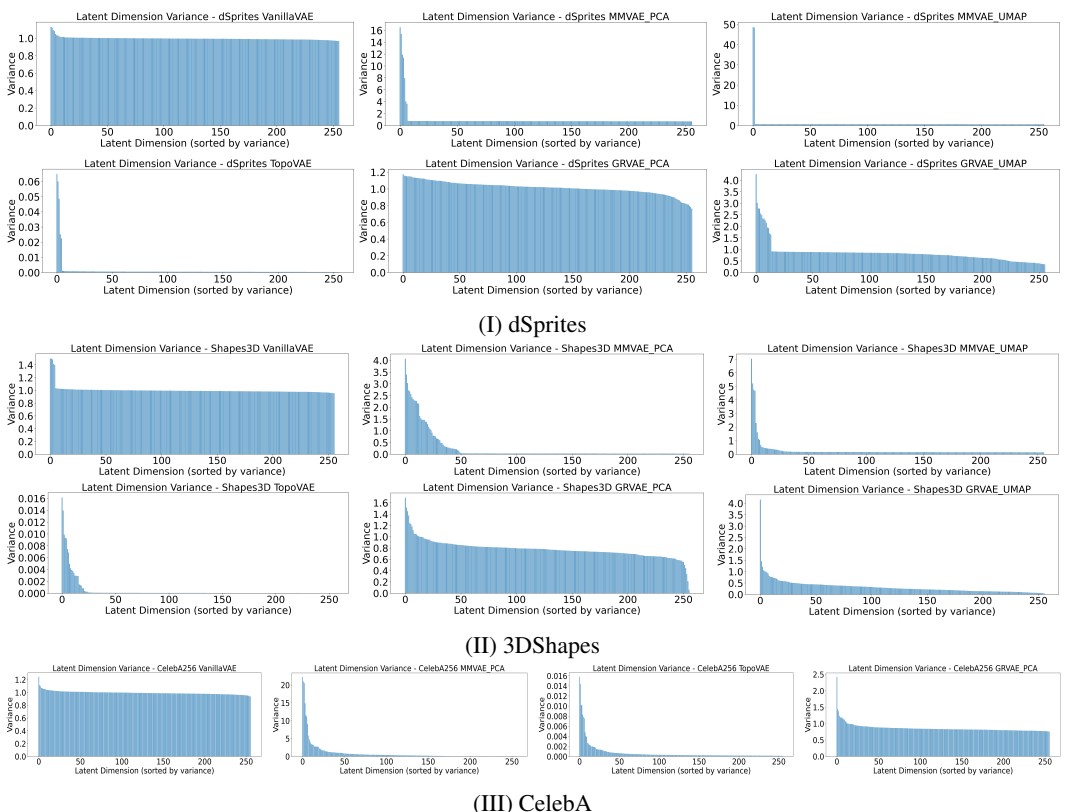

Figure 7: Latent variance per dimension (sorted).

factors. **Limitations:** The primary constraint is the computational cost to calculate low-dimensional projections, particularly for large datasets. For example, we limited CelebA experiments to PCA due to its computational efficiency and superior neighborhood preservation in other datasets. However, projections need only be computed once for a given geometry. For very large datasets, an alternative could be to train using MMLoss on a sufficiently large subset of the data, and then embed the remaining. **Future Directions:** The most promising extension involves applying Manifold Matching to other generative architectures, more suitable for generating high-quality images. For instance, Pandey et al. (2022) use VAE latent spaces as initialization for diffusion processes, but suffer from standard VAE blurriness and lack of meaningful details. Our approach could provide better-structured initializations with better control over meaningful attributes.

# 7 CONCLUSION

We introduced MMAEs, a simple yet effective framework for controlling latent space geometry through distance-based alignment with precomputed references. Our key discovery is that MMAE combined with PCA achieves superior NLDR metrics in large bottleneck scenarios (256D), rivaling sophisticated topological regularizations while maintaining significantly better reconstruction quality and computational efficiency. In generative applications, our approach demonstrates a crucial advantage: the ability to concentrate variance in fewer dimensions while achieving higher absolute variance values. This combination enables superior recovery of semantically meaningful directions—such as changing shape, scale, or orientation in 3DShapes, or adding mustaches and smiles in CelebA faces. Our experiments reveal that optimal isolation of semantically meaningful attributes requires both high absolute variance accumulation and its concentration in fewer dimensions—a property that emerges naturally in our framework. This raises intriguing questions about the synergy between PCA's linear structure and autoencoders' nonlinear capacity. This work provides both theoretical insights into latent space organization and a practical tool for controllable synthetic image generation.

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

# A  APPENDIX

## A.1  PROPERTIES OF THE MANIFOLD-MATCHING LOSS

**Property 1: Scale Invariance** The manifold-matching loss $\mathcal{L}_{mm}$ is invariant to uniform scaling of either the latent space $\mathbf{Z}$ or the reference embedding space $\mathbf{E}$.

Consider uniform scaling of the latent space by factor $\alpha > 0$: $\mathbf{z}'_i = \alpha\mathbf{z}_i$. Then $d_{ij}^{Z'} = \|\alpha\mathbf{z}_i - \alpha\mathbf{z}_j\|_2^2 = \alpha^2\|\mathbf{z}_i - \mathbf{z}_j\|_2^2 = \alpha^2 d_{ij}^Z$. The normalization ensures:

$$\tilde{d}_{ij}^{Z'} = \frac{\alpha^2 d_{ij}^Z}{\|\boldsymbol{\alpha}^2\mathbf{D}^Z\|_F} = \frac{\alpha^2 d_{ij}^Z}{\alpha^2\|\mathbf{D}^Z\|_F} = \tilde{d}_{ij}^Z$$

Therefore, $\mathcal{L}_{mm}$ remains unchanged under uniform scaling.

Scale invariance enables the autoencoder to concentrate absolute variance in fewer dimensions while maintaining distance relationships, as the loss function is insensitive to the magnitude scaling that occurs during variance redistribution across latent dimensions. This property explains why MMVAEs achieve superior variance accumulation compared to GRVAEs, which constrain variance through their respective regularization mechanisms.

**Property 2: Dimensionality Independence** The manifold-matching loss enables meaningful comparison between latent and reference spaces of different dimensionalities ($m \neq l$) through distance preservation rather than pointwise alignment.

This independence arises because the loss operates on pairwise distance matrices $\mathbf{D}^Z \in \mathbb{R}^{p\times p}$ and $\mathbf{D}^E \in \mathbb{R}^{p\times p}$, which have identical dimensions regardless of the original space dimensionalities. For a batch of size $p$, both distance matrices are $p \times p$, whether the latent space $\mathbf{Z}$ has dimension $m = 256$ and the reference embedding $\mathbf{E}$ has dimension $l = 2$ or $l = 100$.

The normalization step $\tilde{d}_{ij} = d_{ij}/\|\mathbf{D}\|_F$ ensures that distance matrices from spaces of different scales become comparable, focusing the optimization on relative geometric relationships rather than absolute magnitudes. This contrasts with point wise alignment methods like GRAE, which require $\mathcal{L}_g = \sum_{i=1}^k \|\boldsymbol{\epsilon}_i - g_\theta(\mathbf{x}_i)\|^2$ and thus demand $m = l$ for meaningful optimization.

## A.2  VARIATIONAL AUTOENCODER EXTENSION

### A.2.1  MMVAE FORMULATION

The manifold-matching principle extends naturally to variational autoencoders (Kingma & Welling, 2022). In standard VAEs, the encoder $q_\phi(\mathbf{z}|\mathbf{x})$ parameterizes a posterior distribution $\mathcal{N}(\boldsymbol{\mu}_\phi(\mathbf{x}), \boldsymbol{\sigma}_\phi^2(\mathbf{x}))$, while the decoder $p_\theta(\mathbf{x}|\mathbf{z})$ models the conditional likelihood. The VAE objective combines reconstruction and KL regularization:

$$\mathcal{L}_{\text{VAE}}(\theta, \phi; \mathbf{x}) = -\mathbb{E}_{q_\phi(\mathbf{z}|\mathbf{x})}[\log p_\theta(\mathbf{x}|\mathbf{z})] + \beta\text{KL}(q_\phi(\mathbf{z}|\mathbf{x})\|p(\mathbf{z})) \tag{7}$$

where $p(\mathbf{z}) = \mathcal{N}(\mathbf{0}, \mathbf{I})$ and $\beta$ controls regularization strength.

Recent work (Chadebec & Allassonnière, 2022) reveals that VAEs implicitly learn geometric structure in the latent space: the learned means $\boldsymbol{\mu}_\phi(\mathbf{x})$ and variances $\boldsymbol{\sigma}_\phi^2(\mathbf{x})$ encode how the model measures uncertainty and relationships between encoded points. Through reconstruction training, the VAE learns that certain spatial arrangements in latent space correspond to meaningful transformations in data space.

For MMVAE, we augment the VAE objective with manifold-matching:

$$\mathcal{L}_{\text{MMVAE}}(\theta, \phi; \mathbf{X}, \mathbf{E}) = \mathcal{L}_{\text{VAE}}(\theta, \phi; \mathbf{X}) + \lambda\mathcal{L}_{mm}(\mathbf{Z}, \mathbf{E}) \tag{8}$$

where $\mathbf{Z} = \{\mathbf{z}_i\}_{i=1}^p$ represents the reparameterized latent variables $\mathbf{z}_i = \boldsymbol{\mu}_\phi(\mathbf{x}_i) + \boldsymbol{\sigma}_\phi(\mathbf{x}_i) \odot \boldsymbol{\epsilon}_i$ with $\boldsymbol{\epsilon}_i \sim \mathcal{N}(\mathbf{0}, \mathbf{I})$.

This approach leverages the VAE's natural tendency to learn meaningful geometric relationships: our distance-based regularization provides explicit guidance for this geometric learning process by aligning it with known good structures from reference embeddings $\mathbf{E}$. This explains the superior performance of MMVAE—rather than fighting against the VAE's geometric learning, we direct it toward beneficial configurations.

### A.2.2 TRAINING PROCEDURE

---

**Algorithm 2** MMVAE Training Step

---

**Require:** Batch $\mathbf{X}_b$, embeddings $\mathbf{E}_b$, encoder $q_\phi$, decoder $p_\theta$
1: Compute posterior: $\boldsymbol{\mu}_b, \log \boldsymbol{\sigma}_b^2 = q_\phi(\mathbf{X}_b)$
2: Sample: $\mathbf{z}_b = \boldsymbol{\mu}_b + \boldsymbol{\sigma}_b \odot \boldsymbol{\epsilon}$ where $\boldsymbol{\epsilon} \sim \mathcal{N}(\mathbf{0}, \mathbf{I})$
3: Reconstruct: $\hat{\mathbf{X}}_b = p_\theta(\mathbf{z}_b)$
4: Compute losses: $\mathcal{L}_r, \mathcal{L}_{KL}, \mathcal{L}_{mm}(\boldsymbol{\mu}_b, \mathbf{E}_b)$
5: Total: $\mathcal{L} = \mathcal{L}_r + \beta\mathcal{L}_{KL} + \lambda\mathcal{L}_{mm}$
6: Update: $\theta, \phi \leftarrow \text{Adam}(\nabla_{\theta,\phi}\mathcal{L})$

---

### A.3 TRAINING & ARCHITECTURE SPECIFICATIONS

This section provides detailed specifications for all architectures used across our experiments. Our implementation uses PyTorch (Paszke et al., 2019) for neural network construction and training. The manifold learning components use scikit-learn (Pedregosa et al., 2011) for PCA implementation and t-SNE, and UMAP-learn (McInnes et al., 2018) for UMAP embeddings.

### A.3.1 DEEPAE ARCHITECTURE

Used for MNIST, Fashion-MNIST, and CIFAR-10 datasets in both 2D visualization and NLDR quality experiments.

Table 1: DeepAE Architecture Specifications

| Layer | Input Size | Output Size | Activation | Notes |
|---|---|---|---|---|
| **Encoder** | | | | |
| Linear | input_dim | 1000 | - | Flattened input |
| BatchNorm1d | 1000 | 1000 | - | |
| ReLU | 1000 | 1000 | ReLU | |
| Linear | 1000 | 500 | - | |
| BatchNorm1d | 500 | 500 | - | |
| ReLU | 500 | 500 | ReLU | |
| Linear | 500 | 250 | - | |
| BatchNorm1d | 250 | 250 | - | |
| ReLU | 250 | 250 | ReLU | |
| Linear | 250 | latent_dim | - | Bottleneck |
| **Decoder** | | | | |
| Linear | latent_dim | 250 | - | |
| BatchNorm1d | 250 | 250 | - | |
| ReLU | 250 | 250 | ReLU | |
| Linear | 250 | 500 | - | |
| BatchNorm1d | 500 | 500 | - | |
| ReLU | 500 | 500 | ReLU | |
| Linear | 500 | 1000 | - | |
| BatchNorm1d | 1000 | 1000 | - | |
| ReLU | 1000 | 1000 | ReLU | |
| Linear | 1000 | input_dim | Tanh | Output reconstruction |

### A.3.2 VAE ARCHITECTURES

### A.4 EXTENDED EXPERIMENTAL RESULTS

### A.4.1 CLASSIFICATION AND CLUSTERING

The latent spaces of trained autoencoders can be evaluated on downstream tasks such as clustering and classification. We assess the learned representations using a single-layer MLP classifier for clas-

Table 2: CelebA Convolutional VAE Architecture

| Layer | Input Size | Output Size | Kernel/Stride | Activation |
|---|---|---|---|---|
| **Encoder** | | | | |
| Conv2d | 3×256×256 | 32×256×256 | 3×3/1, pad=1 | ReLU + BatchNorm |
| Conv2d | 32×256×256 | 64×128×128 | 4×4/2, pad=1 | ReLU + BatchNorm |
| Conv2d | 64×128×128 | 128×64×64 | 4×4/2, pad=1 | ReLU + BatchNorm |
| Conv2d | 128×64×64 | 256×32×32 | 4×4/2, pad=1 | ReLU + BatchNorm |
| Conv2d | 256×32×32 | 512×16×16 | 4×4/2, pad=1 | ReLU + BatchNorm |
| Conv2d | 512×16×16 | 1024×8×8 | 4×4/2, pad=1 | ReLU + BatchNorm |
| Conv2d | 1024×8×8 | 2048×4×4 | 4×4/2, pad=1 | ReLU + BatchNorm |
| Conv2d | 2048×4×4 | 4096×1×1 | 4×4/1, pad=0 | - |
| Flatten | 4096×1×1 | 4096 | - | - |
| $\text{Linear}_\mu$ | 4096 | latent_dim | - | - (VAE mean) |
| $\text{Linear}_{\log \sigma^2}$ | 4096 | latent_dim | - | - (VAE log variance) |
| **Decoder** | | | | |
| Linear | latent_dim | 4096 | - | - |
| Reshape | 4096 | 4096×1×1 | - | - |
| ConvTranspose2d | 4096×1×1 | 2048×4×4 | 4×4/1, pad=0 | ReLU + BatchNorm |
| ConvTranspose2d | 2048×4×4 | 1024×8×8 | 4×4/2, pad=1 | ReLU + BatchNorm |
| ConvTranspose2d | 1024×8×8 | 512×16×16 | 4×4/2, pad=1 | ReLU + BatchNorm |
| ConvTranspose2d | 512×16×16 | 256×32×32 | 4×4/2, pad=1 | ReLU + BatchNorm |
| ConvTranspose2d | 256×32×32 | 128×64×64 | 4×4/2, pad=1 | ReLU + BatchNorm |
| ConvTranspose2d | 128×64×64 | 64×128×128 | 4×4/2, pad=1 | ReLU + BatchNorm |
| ConvTranspose2d | 64×128×128 | 32×256×256 | 4×4/2, pad=1 | ReLU + BatchNorm |
| ConvTranspose2d | 32×256×256 | 3×256×256 | 3×3/1, pad=1 | Tanh |

Table 3: dSprites VAE Architecture Specifications

| Layer | Input Size | Output Size | Kernel/Stride | Activation |
|---|---|---|---|---|
| **Encoder** | | | | |
| Conv2d | 1×64×64 | 32×32×32 | 4×4/2, pad=1 | LeakyReLU(0.2) |
| Conv2d | 32×32×32 | 64×16×16 | 4×4/2, pad=1 | LeakyReLU(0.2) + BatchNorm |
| Conv2d | 64×16×16 | 128×8×8 | 4×4/2, pad=1 | LeakyReLU(0.2) + BatchNorm |
| Conv2d | 128×8×8 | 256×4×4 | 4×4/2, pad=1 | LeakyReLU(0.2) + BatchNorm |
| Flatten | 256×4×4 | 4096 | - | - |
| Linear | 4096 | 512 | - | LeakyReLU(0.2) |
| $\text{Linear}_\mu$ | 512 | latent_dim | - | - (VAE mean) |
| $\text{Linear}_{\log \sigma^2}$ | 512 | latent_dim | - | - (VAE log variance) |
| **Decoder** | | | | |
| Linear | latent_dim | 512 | - | LeakyReLU(0.2) |
| Linear | 512 | 4096 | - | LeakyReLU(0.2) |
| Reshape | 4096 | 256×4×4 | - | - |
| ConvTranspose2d | 256×4×4 | 128×8×8 | 4×4/2, pad=1 | LeakyReLU(0.2) + BatchNorm |
| ConvTranspose2d | 128×8×8 | 64×16×16 | 4×4/2, pad=1 | LeakyReLU(0.2) + BatchNorm |
| ConvTranspose2d | 64×16×16 | 32×32×32 | 4×4/2, pad=1 | LeakyReLU(0.2) + BatchNorm |
| ConvTranspose2d | 32×32×32 | 1×64×64 | 4×4/2, pad=1 | Tanh |

sification performance and compute silhouette scores and Adjusted Rand Index (ARI) for clustering evaluation on MNIST, Fashion-MNIST, and CIFAR-10 datasets.

Our results show that MMAEs achieve comparable to slightly superior performance relative to standard (vanilla) autoencoders while maintaining similar levels of global structure preservation measure by RMSE. In contrast, TopoAE demonstrates the poorest classification and clustering performance across all metrics. The authors of TopoAE (Moor et al., 2020b) acknowledge that topological preservation can prove challenging for classification tasks, arguing that the goal of increasing class separability may conflict with preserving topological structures.

Table 4: 3DShapes VAE Architecture Specifications

| Layer | Input Size | Output Size | Kernel/Stride | Activation |
|---|---|---|---|---|
| **Encoder** | | | | |
| Conv2d | 3×64×64 | 32×32×32 | 4×4/2, pad=1 | LeakyReLU(0.2) |
| Conv2d | 32×32×32 | 64×16×16 | 4×4/2, pad=1 | LeakyReLU(0.2) + BatchNorm |
| Conv2d | 64×16×16 | 128×8×8 | 4×4/2, pad=1 | LeakyReLU(0.2) + BatchNorm |
| Conv2d | 128×8×8 | 256×4×4 | 4×4/2, pad=1 | LeakyReLU(0.2) + BatchNorm |
| Conv2d | 256×4×4 | 512×2×2 | 4×4/2, pad=1 | LeakyReLU(0.2) + BatchNorm |
| Flatten | 512×2×2 | 2048 | - | - |
| Linear | 2048 | 1024 | - | LeakyReLU(0.2) + Dropout(0.2) |
| Linear | 1024 | 512 | - | LeakyReLU(0.2) |
| Linear$_\mu$ | 512 | latent_dim | - | - (VAE mean) |
| Linear$_{\log \sigma^2}$ | 512 | latent_dim | - | - (VAE log variance) |
| **Decoder** | | | | |
| Linear | latent_dim | 512 | - | LeakyReLU(0.2) |
| Linear | 512 | 1024 | - | LeakyReLU(0.2) + Dropout(0.2) |
| Linear | 1024 | 2048 | - | LeakyReLU(0.2) |
| Reshape | 2048 | 512×2×2 | - | - |
| ConvTranspose2d | 512×2×2 | 256×4×4 | 4×4/2, pad=1 | LeakyReLU(0.2) + BatchNorm |
| ConvTranspose2d | 256×4×4 | 128×8×8 | 4×4/2, pad=1 | LeakyReLU(0.2) + BatchNorm |
| ConvTranspose2d | 128×8×8 | 64×16×16 | 4×4/2, pad=1 | LeakyReLU(0.2) + BatchNorm |
| ConvTranspose2d | 64×16×16 | 32×32×32 | 4×4/2, pad=1 | LeakyReLU(0.2) + BatchNorm |
| ConvTranspose2d | 32×32×32 | 3×64×64 | 4×4/2, pad=1 | Tanh |

However, our findings suggest it is possible to preserve the original data topology while maintaining or improving classification accuracy, as demonstrated in Figure 8. This indicates that geometric regularization through manifold-matching does not necessarily compromise the utility of learned representations for discriminative tasks.

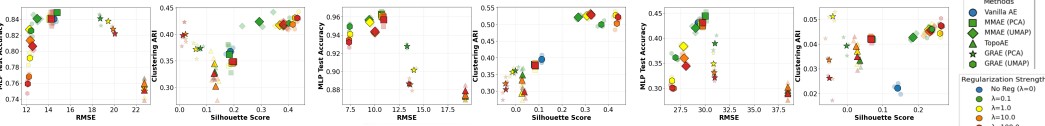

Figure 8: Classification versus RMSE, and Clustering ARI vs Silhouette score for varying $\lambda$ strength on MNIST, F-MNIST, and CIFAR10.

### A.4.2 CORRUPTED IMAGES

We evaluate the preservation of original data topology under three types of corruption: Gaussian noise, Gaussian blur, and brightness changes. Figure 9 shows average results across 10 runs for neighborhood preservation (trustworthiness × continuity), RMSE, and MRRE metrics.

Neighborhood preservation and MRRE results show clear performance distinctions between models on corrupted data. All regularized approaches outperform standard autoencoders, with performance correlating with regularization strength $\lambda$—higher values achieve better robustness. MMAE-PCA and TopoAE, which demonstrate the highest preservation on clean data, maintain superior performance under corruption. MRRE consistently shows that stronger regularization achieves lower error across all corruption types.

RMSE anomaly: Unexpectedly, RMSE values drop significantly for all models when moving from clean to corrupted data, with minimal differences between approaches. This contrasts sharply with the clear model distinctions observed in other metrics. We hypothesize this occurs because corruption creates more uniform distance distributions in both input and latent spaces. When corruption affects all data points similarly (e.g., adding uniform noise), it compresses the dynamic range of pairwise distances, making distance matrices more homogeneous. This artificial similarity between

corrupted input and latent distance matrices leads to lower RMSE values, even though actual topology preservation may be degraded.

This suggests that RMSE becomes less discriminative under corruption, while MRRE and neighborhood preservation metrics remain reliable indicators of model robustness to noise.

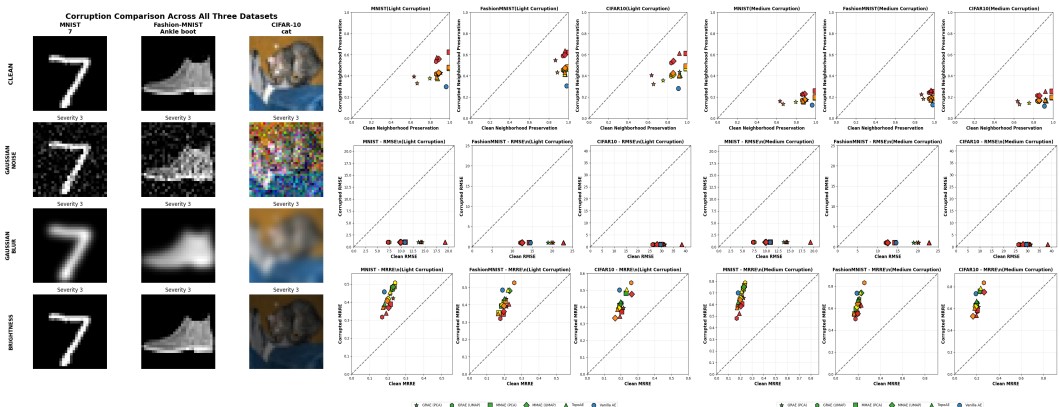

Figure 9: Examples of medium corruption (Severity 3) (Left). Corrupted vs clean NLDR metrics (Right).

### A.4.3 REGULARIZATION STRENGTH ANALYSIS

The regularization strength $\lambda$ controls the balance between reconstruction fidelity and geometric structure preservation. Lower values close to 0 give little weight to resemblance to the reference embeddings, while higher values more rigorously bind the autoencoder latent space to the reference. Figure 10 shows results starting with $\lambda = 1$ at epoch 1 and reducing by a factor of 0.1 every 20 epochs over 200 epochs total. Even when the strength is significantly reduced, some traits from the reference are maintained.

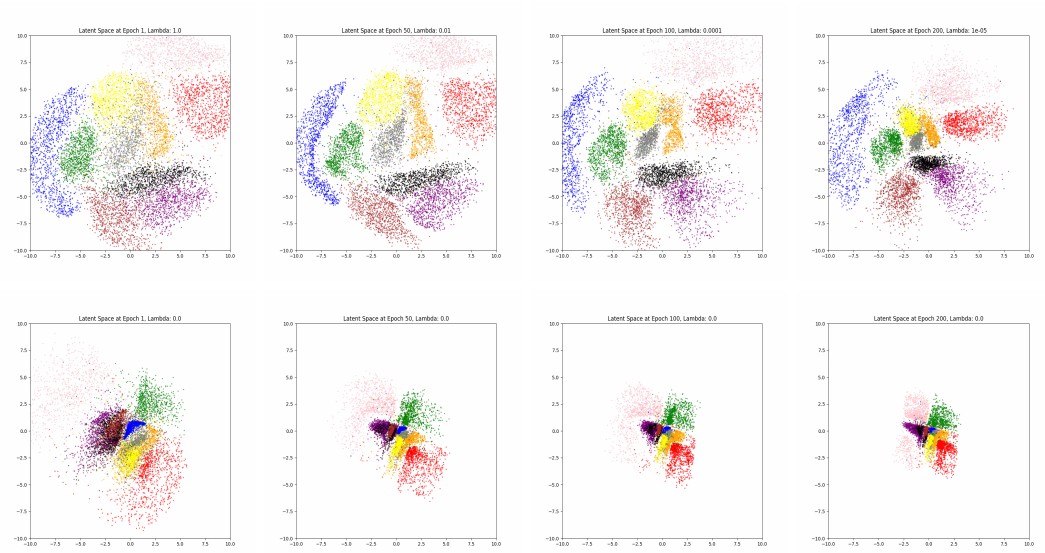

Figure 10: Comparison of latent spaces between MMAE (top row) and AE (bottom row) in MNIST. Decreasing $\lambda$ by a factor of 0.1 every 20 epochs, 200 total.

### A.4.4  UMAP/T-SNE HYPERPARAMETER VARIATIONS

UMAP and t-SNE have hyperparameters that can significantly alter the final embedding appearance. These are choices to be made for each use case, and in summary, it doesn't affect the training procedure as MMLoss only requires a valid distance matrix to operate. It is thus flexible to the choise of hyperparameters making it a general solution to extrapolate known representations, as can be seen in Figure 11.

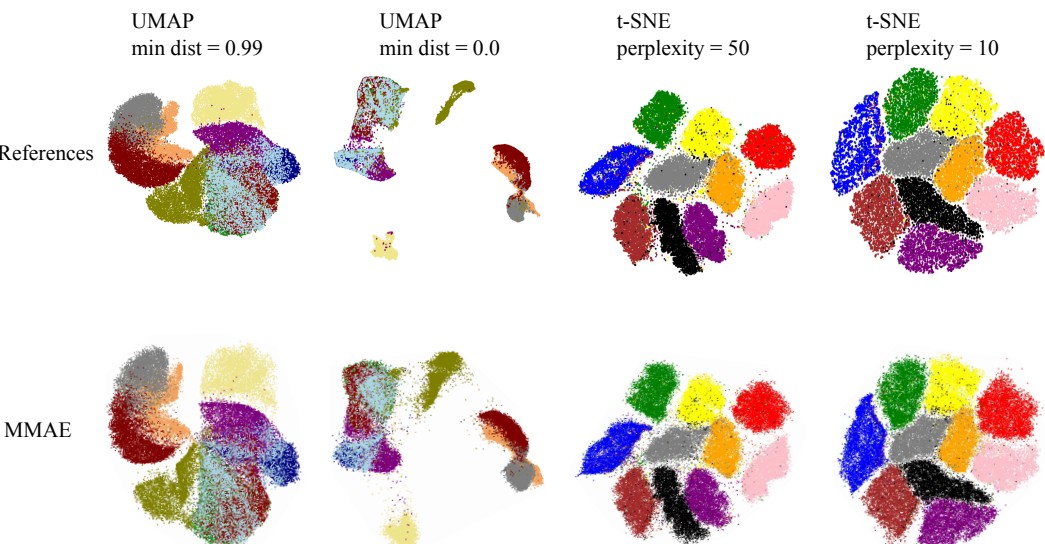

Figure 11: MMAEs copying embeddings for the MNIST and F-MNIST dataset under different hyperparameter combinations.

