# OpenReview forum: "Manifold-Matching Autoencoders"
_ICLR.cc/2026/Conference — ICLR 2026 Conference Withdrawn Submission_

### Official Review · Reviewer_VzYg · 2025-10-27

**Soundness:** 2
**Presentation:** 1
**Contribution:** 1
**Rating:** 2
**Confidence:** 5

**Summary:**

The paper introduces a Manifold-Matching Autoencoders (MMAEs), a framework that aligns autoencoder latent spaces with precomputed geometric references. This is accomplished by using distance-based regularization to match latent and reference distance matrices.

**Strengths:**

The paper's organization is mostly fine, the main ideas are easy to follow.

**Weaknesses:**

1) The main weakness is a limited scientific novelty. In a nutshell, authors propose to add a distance-based regularizer between real-latent spaces to an AE objective.

2) Experimental results in Fig. 3 are very hard to interpret. Consider presenting results in a Table or by a different visualization.
Comparison with SOTA methods like IVIS, RTD-AE, PacMAP, PHATE, and MDS are missing.
Consider evaluating linear correlation, the triplet distance ranking accuracy, Wasserstein distance between persistence barcodes (Moor et al., 2020b; Trofimov et al., 2023)

3) Figure 4 is hard to interpret. Consider using quantitative evaluation, maybe one can train a classifier for detecting quality of interpolation or use human assessment by services like Mechanical Turk.
A comparison with SOTA disentanglement methods like DAVE, beta-VAE, beta-TCVAE, FactorVAE are missing.
Consider evaluating standard disentanglement metrics like MIG, FactorVAE score MIG, SAP, DCI.

**Questions:**

1) How do you select a regularization coefficient lambda?
2) What is your interpretation of experimental results in Fig. 1? It seems to be no significant difference after addition of MM regularizer.

---

### Official Review · Reviewer_jyBR · 2025-10-30

**Soundness:** 2
**Presentation:** 3
**Contribution:** 2
**Rating:** 4
**Confidence:** 5

**Summary:**

The paper introduces a new dimensionality reduction method called Manifold-Matching Autoencoder (MMAE). This is essentially a parametrized version of a well-known Multidimensional Scaling (MDS), where the pairwise distances are obtained from precomputed embeddings, for example, using PCA. As in MDS, the loss is calculated between the normalized pairwise distance matrices and added to the standard reconstruction loss. The authors report experiments in two scenarios -- visualization and generation (MMVAE).

**Strengths:**

- **Text**: The paper is nicely structured and easy to follow. The idea is explained in detail.
 - **Experiments**: two setups are explored -- generative (VAE) and NLDR.

**Weaknesses:**

- **Idea**: The method depends entirely on precomputed embeddings (e.g., PCA, UMAP), which already provide a geometric structure. It is not clearly explained why training a parametric autoencoder is required beyond simply using these embeddings directly. The paper argues that topological autoencoder methods fail in higher-dimensional bottlenecks due to the computational costs of persistence homology; however, just like MMAE, it also operates on the pairwise distances and doesn't depend on the dimension, so this claim is questionable and not supported by the experimental evidence.
 - **Metrics and results**: RMSE between distance matrices appears identical to the training loss (Eq. 3). Reporting it as a performance metric risks circular reasoning and does not reflect true global structure preservation. To substantiate claims of global and local structure preservation, additional metrics should be included, like clustering quality (ARI, Silhouette) for local structure, and global topology metrics from TopoAE or RTD-AE. Figure 6 (II) shows that MRRE (which measures rank correlation of pairwise distances) is lower for MMVAE than for TopoVAE on the reported datasets, contradicting the claim that MMVAE best preserves global structure. The authors do not discuss these inconsistencies. The claim that MMAE captures global topology is unsupported without evaluation on synthetic manifolds with known structure (e.g., spheres, Swiss rolls, rings). Current datasets (MNIST, F-MNIST, CIFAR-10, CelebA) have complex, unknown topology, so qualitative “global structure preservation” is difficult to verify. Many improvements reported for “global structure” may derive from alignment with PCA rather than genuine topology preservation. Without baseline comparisons to raw PCA or UMAP embeddings in the results tables, it is hard to isolate the effect of the proposed regularization.

**Questions:**

- How exactly is the RMSE score computed? Why do lower MRRE values indicate better performance, and how should we interpret the discrepancies where TopoVAE achieves lower MRRE than MMVAE (Figure 6 (II))?
 - To substantiate the claim of global structure preservation, could you provide results on datasets with known topology (e.g., Spheres & rings from TopoAE)?

---

### Official Review · Reviewer_ECfY · 2025-10-30

**Soundness:** 3
**Presentation:** 2
**Contribution:** 2
**Rating:** 2
**Confidence:** 4

**Summary:**

This paper proposes to enhance an autoencoder by aligning the latent space with precomputed reference embeddings. In particular, the alignment is done through matching the pairwise distances of the latent vectors with those of the reference embeddings. The authors argue that this approach better preserves the topological structure of high-dimensional data.

**Strengths:**

This paper originates from a good motivation: preserving the data topology in the latent space. Topological data analysis (TDA) approaches are either too expensive in computation or too narrowly applicable (e.g., to only 2D/3D visualization). The proposed method addresses the drawbacks of TDA by regularizing an autoencoder, a simple and elegant approach. However, despite its simplicity, the paper has not done enough theoretical/empirical exploration (see the Weakness section below).

The authors touch on the concept of variance concentration/hierarchy, which appears to be a good merit of the proposed method. However, the exposition of this concept is insufficient (also see the Weakness section below).

**Weaknesses:**

Despite a good motivation, the paper leaves more to be desired.

The theoretical foundation is weak. The proposed distance regularization looks ad hoc. It is unclear why the authors propose such a regularization among many obvious alternatives.

It might be helpful if the authors could set the stage with more background information on topology/manifold and topological data analysis, as justification for the proposed method.

A weakness of the method is that it leaves open what a good reference embedding is. The proposal is more like a framework, which can be applied to any reference embedding, but it is unclear what good references are.

The authors should put more exposition on variance concentration: what it is, why it is a good thing, why the method leads to it, etc.

The experiment results are not compelling. For example, in Figure 4, the interpolation quality for 3DShapes is poor. The smallest balls for methods (b), (c), and (f) and the largest balls for methods (a), (d), and (e) have significant distortions.

The fonts in the figures are too small to be legible.

For CIFAR10/MNIST/F-MNIST datasets, the authors use overly simple neural networks. They should at least try convolution layers for images. It is suspected that the neural networks, not just the training loss, are key to the quality of Figures 4 and 5.

**Questions:**

See the Weakness section.

---

### Official Review · Reviewer_qnQE · 2025-11-03

**Soundness:** 2
**Presentation:** 1
**Contribution:** 1
**Rating:** 2
**Confidence:** 4

**Summary:**

The paper proposes Manifold-Matching Autoencoders (MMAE), which learn latent representations by matching pairwise distances to a reference embedding (typically produced by a manifold learning method). The quality of dimensionality reduction is evaluated on high-dimensional datasets and claimed to outperform methods such as Topological Autoencoders (TopoAE) and Geometry-Regularized Autoencoders (GAE). Combined with a VAE objective, the model is claimed to support semantic interpolation even in high-dimensional latent spaces. The paper also reports a concentration phenomenon in latent-space variance and posits this as a driver of improved interpolation.

**Strengths:**

- The approach is scale-invariant and does not require the latent dimensionality to equal the intrinsic manifold dimension, potentially simplifying autoencoder design and enabling high-dimensional latents.
- It is intriguing that geometry preservation appears to improve semantic interpolation quality, although the underlying rationale remains unclear.

**Weaknesses:**

- **Limited novelty.** Similar ideas that preserve the geometry of embeddings/graphs already exist. Direct comparisons are missing, e.g.
  - SPAE: Singh & Nag (2021), Structure-preserving deep autoencoder-based dimensionality reduction for data visualization, IEEE SNPD.
  - GGAE: Lim, Kim, Lee, Jang & Park (2024), Graph Geometry-Preserving Autoencoders, ICML.

  The paper needs to position MMAE against these and related geometry/graph-preserving autoencoders with clear conceptual and empirical distinctions. To my knowledge, the proposed regularizer is conceptually very similar to SPAE when formulated to preserve reference distances.
- **Clarity and reproducibility.** Key experimental details are unspecified or under-specified, including:
  -	How reference distance matrices are constructed (PCA/UMAP settings).
  -	Train/validation/test splits.
  -	Statistical reporting (confidence intervals).
- **Empirical significance.** From Figures 5–6, superiority over baselines is not compelling; differences appear small and sometimes ambiguous. Stronger baselines (e.g., SPAE, GGAE) and ablations (e.g., latent-dim sensitivity) are needed.
- **Unconvincing explanation of variance concentration.** The reported concentration in latent variance is not theoretically justified, nor is its causal link to improved semantic interpolation established.
- **Positioning vs. graph/geometry-preserving objectives.** It remains unclear why MMAE should outperform graph- or geometry-preserving autoencoders when using the same reference distances. If the advantage arises from the VAE regularizer, the latent dimensionality, or the scale-invariance property of the loss, this should be clarified and verified through ablation studies.

**Questions:**

- Please refer to the points raised in the weaknesses section.
- Why can MMAE perform better than GAE even under identical reference distance matrices? A thorough analysis and ablation study of this point would be helpful.
- How does MMAE compare to SPAE? What are the possible advantages and disadvantages of the obtained latent representations?

---

### Note · Authors · 2026-01-22

I have read and agree with the venue's withdrawal policy on behalf of myself and my co-authors.